# Barriers to implementing patient safety incident reporting and learning guidelines in specialised care units, KwaZulu-Natal: A qualitative study

**T. M. H. Gqaleni©\*, Sipho W. Mkhize**

School of Nursing and Public Health, College of Health Sciences, University of KwaZulu-Natal, Durban, South Africa

\* gqalenit@ukzn.ac.za

## Abstract

### Background

Globally, increased occurrences of patient safety incidents have become a public concern. The implementation of Patient safety incidents reporting and learning guidelines is fundamental to reducing preventable patient harm. To improve the implementation of these guidelines in specialised care units in KwaZulu-Natal, the views of healthcare professionals were unearthed.

### Aim

This study explores the healthcare professionals' views toward the implementation of Patient safety incident reporting and learning guidelines in specialised care units.

### Methods

A descriptive, explorative qualitative approach was used to collect qualitative data from healthcare professionals working in specialised care units. The study was conducted in specialised care units of three purposely selected public hospitals in two districts of KwaZulu-Natal. Group discussions and semi-structured interviews were conducted from August to October 2021. Content data analysis was performed using Tesch's method of analysis process.

### Results

The main themes that emerged during data analysis were; ineffective reporting system affecting the communication of Patient safety incident guidelines, inadequate institutional and management support for the healthcare professionals, insufficient education and training of healthcare professionals, and poor human resources affecting the implementation of Patient Safety Incident guidelines. The findings highlighted that there were more major barriers to the implementation of the Patient safety incident reporting and learning guidelines.

**Data Availability Statement:** All relevant data are within the manuscript and minimal data set in Supporting Information files.

**Funding:** Author: TMH Gqaleni Grant Number:129904 Name of the Funder: National Research Foundation ERL: https://nrfsubmission.nrf.ac.za The funders had no role in study design, data collection and analysis, decision to publish, or preparation of the manuscript.

**Competing interests:** The authors have declared that no competing interests exist.

## Conclusion

This study confirmed that the Patient safety incident reporting and learning guidelines are still not successfully implemented in the specialised care units and the barriers to implementation were highlighted. For rigorous implementation in South Africa, the study recommends revised Patient safety incident reporting and learning guidelines, designed in consultation with the frontline healthcare professionals. These must consist of standardised, simple-user-friendly reporting process as well as a better implementation strategy to guide the healthcare professionals. Continuous professional development programmes may play an important role in the facilitation of the implementation process.

## Introduction

Globally, increased occurrence of Patient safety incidents (PSIs) and near misses have become a public concern, which includes South Africa [1]. Near misses are incidents or situations that have the potential to cause harm but do not reach the patient due to timely interventions, whereas a PSI is harm caused by medical mismanagement, instead of the underlying disease [2]. Unsafe care is one of the top 10 leading causes of death worldwide and up to 83% of harm is avoidable [3]. The Institute of Medicine reveals deaths related to PSIs vary from 44 000 to 98 000 people each year with high-cost implications [4–6]. According to Flott, Fontana [3], in 2013, over 420 million hospitalisations around the world resulted in nearly 43 million PSIs. An analysis by the Organisation for Economic Co-operation and Development (OECD) found that 15% of all hospital costs in OECD nations are due to patient harm from PSIs, with associated costs ranging between $1.4 to $1.6 trillion, each year [7]. According to Choi Pyo, et.al, improving the hospital organization atmosphere is essential to facilitate useful communication of patient safety incidents [8, 9].

In Swedish and Brazilian health services, barriers and facilitators that influenced patient safety were identified and the relevance of a multifaceted system perspective on patient safety problems and solutions was pointed out [10, 11]. Patient safety is a multifactorial concept, driven by a complicated array of technical, human and system factors [3, 9]. Several studies have identified barriers to the implementation of PSIs reporting in healthcare settings including specialised care units (SCUs). These include underreporting, punitive culture, lack of standardized reporting system, staff shortages, lack of safety culture, poor processing of incident reports, inadequate engagement of healthcare professionals, limited institutional support of incident reporting systems including inadequate usage of evolving health information technology [1, 12–17].

In middle and low-income countries (LMICs), almost one-third of patients who suffered PSIs died and four out of five of those incidences were preventable [18]. Moreover, 134 million PSIs occur in hospitals due to unsafe care, resulting in 2.6 million deaths, each year (National Academies of Sciences and Medicine, 2018). Also, it is estimated that 75% of PSIs occur in the LMIC contexts, during hospital care and 10–15% of total deaths are as a result of poor quality care [3]. In South Africa, a study conducted in KwaZulu-Natal reveals a high percentage of serious PSIs (47%) indicating the poor implementation of PSI reporting guidelines and lack of improvement strategies [19]. While incident reporting is considered a cornerstone of patient safety culture and quality improvement in healthcare [20, 21], under-reporting in hospitals worldwide is still a challenge [1, 22, 23].

In South Africa, the implementation of the PSI reporting and learning systems guidelines, as recommended by the KwaZulu-Natal Department of Health (KZN DoH), in line with the National Department of Health (NDoH 2017) has provided no clear evidence of the reduction of PSIs and if these guidelines are adequately implemented in SCUs. Gqaleni and Mkhize [17], affirmed that healthcare professionals' perception of knowledge of PSI reporting and learning guidelines was good, however, the perception of the implementation was poor. This study further recommended a better implementation strategy to improve the quality and safety of healthcare in SCUs. For SCUs, to heighten the positive performance and practices of PSI reporting and learning guidelines, the views of healthcare professionals needed to be unearthed. Therefore, the study aimed to explore and describe the views of healthcare professionals regarding the implementation of PSI reporting and learning guidelines in specialised care units in KwaZulu-Natal.

## Materials and methods

### Study design and setting

A descriptive, explorative qualitative approach was used to collect qualitative data from healthcare professionals working in SCUs. The study was conducted in SCUs of three purposely selected public hospitals in two districts of KwaZulu-Natal. Two hospitals (A and C), situated in the eThekwini district, provide both secondary and tertiary services, and the third tertiary hospital (B) is in the uMgungundlovu district. The settings consisted of SCUs that included both Critical care units (CCUs), where the most critically ill patients were admitted as well as High care units, where the recovering critically ill patients were transferred but still needed specialised care.

### Population, recruitment, and sampling

Sampling is a process of selecting a portion of the population to represent the entire population so that inferences about the population can be made [24]. The target population for this study consisted of healthcare professionals, who were involved in the implementation of PSI reporting and learning guidelines in the SCUs. Therefore, the participants that were purposely selected had expertise and experience in SCU environments and had vast experience in the handling of PSI reporting and learning guidelines. The focus group discussion sample comprised of Operational nurse managers. Individual interviews included Assistant nurse managers, Monitoring, and evaluation managers, and Consultant medical doctors.

### Inclusion and exclusion criteria

The healthcare workers that participated in this study included Operational nurse managers, Assistant nurse managers, Consultant medical doctors, Monitoring and evaluating managers, who had more than ten years of SCU experience were included in the study. Healthcare workers who did not work in these SCUs and those unwilling to participate in the study were excluded.

### Recruitment strategy

Participants were identified and were purposely selected to participate in the study. Appointments were made by the researcher with the institutions' Senior nurse managers, Operational nurse managers, and all the participants prior to data collection. The researcher had a briefing session with the participants before data collection and explained the aim of the study and its importance to the research. Participants were informed that they were not forced to participate

and that they had a right to withdraw from the study even if they had already given consent, without any negative consequences, thus ensuring voluntary participation.

## Data collection process

This study was conducted during COVID-19 and the researcher followed the institutions' protocols. The data were collected virtually to ensure social distancing, which was reinforced in the boardrooms, including the use of personal protective equipment by the participants. In this study, Skype for individual interviews was used and Zoom was used for the focus group discussions, to collect qualitative data. To ensure privacy, for example, for Skype, participants needed to sign into the interview individually, which prevented unwanted intruders, and for Zoom meetings, passwords were used and the waiting room, which allowed the host researcher to control who entered the video conference. Data was collected for 3 months, from August to October 2021. Information guidelines guided the focus group discussions and individual interviews, which consisted of topics and probing questions to elicit more detailed information from the participants. The researcher prepared one essential question with a list of probing open-ended questions, although the participants were encouraged to speak freely about the topic, to obtain all the information required.

Focus group discussions (FGDs), which consisted of Operational nurse managers (OMs), working in the selected settings, formed homogeneous participants. These participants were also at the forefront of implementing the PSI reporting and learning guidelines. Individual interviews (IDI) consisted of participants who were heterogeneous, namely, Assistant nurse managers (ANMs), Monitoring and evaluation managers (M&E), and Consultant medical doctors (CMD) to solicit more in-depth information. The probing questions were influenced by the participants' responses and the interviewer sought clarity to minimize bias and subjectivity. Data were collected until saturation was reached, that is, no new information emerged. The actual sample size of participants was 41 and data saturation was reached at participant 18. Interviews were conducted and lasted between 50 minutes to 60 minutes, and this allowed the participants to freely elaborate on their opinions on the phenomenon. Participants gave consent to record all the interviews and the researcher transcribed the data verbatim. The data analysis document was stored in the researcher's personal computer which is password-protected. Back-up copies were stored in a password-protected memory stick. All data was only accessible to the researcher, research assistant, and the supervisor.

## Ethical considerations

The University of KwaZulu-Natal's Human Science Research Ethics Committee (HSSREC/ 00001651/2020), and the Directorate: Health Research and Knowledge Management Unit (NHRD Ref: Kz_202010_0240), granted ethical clearance and permission to conduct the study before data collection. Gatekeepers' permission was also obtained from the Chief executive officers, Heads of department, and Senior nurse managers. Grove, Burns, et al., [25] asserted that the researcher must comply with three principles as stated by the Belmont report, namely, the principle of beneficence, the principle of human dignity, and justice, and this was observed throughout the study. Information sheets and informed consent forms were emailed to participants. Participants had to sign informed consent which was emailed back to the researcher, before the data collection. Furthermore, verbal consent to participate and to the recording of the interview voices were obtained from the participants before data collection. Moreover, a record of verbal permission is on the researcher's recording device. This was further recorded on the interview guides, consisting of the participants' site code, date, start, and end of the

interview. The participants were informed of their rights to withdraw from the study at any time, without any negative repercussions.

## Trustworthiness

During the research process, the researcher applied four criteria, to determine trustworthiness, namely, credibility, dependability, confirmability, and transferability [26].

**Credibility.** According to Polit and Beck [27], credibility refers to the confidence in the truth of the data and its interpretations, that is, the data must reveal what it is searching for. The researcher ensured that this study was conducted authentically and the researcher had a prolonged engagement with the participants, during the data collection process, which assisted in gaining an in-depth understating of the phenomena under study. The interviews were conducted between 50 to 60 minutes which allowed enough time for participants to elaborate on their opinions on the phenomenon.

**Dependability.** In this study, the researcher provided detailed information on the research process, findings, and recommendations, so that the other prospective researchers could replicate the study to their context. According to Polit and Beck [27], the findings of an inquiry must be able to be repeated and replicated with the same participants in the same context.

**Confirmability.** Confirmability refers to objectivity, that is, the potential for congruence between two or more independent people about data's accuracy, relevance, or meaning [27]. In this study, the research report conveyed participants' experiences and not the perspectives of the researcher. The themes generated during data collection reflected the tone of the participants. After the data transcription, participants were contacted telephonically, to confirm that the information provided was captured accurately. Also, the supervisor concurred with what was transcribed.

**Transferability.** Transferability refers to the extent to which findings can be generalized and applied in other settings or groups [27]. In this study, the researcher was responsible for providing a detailed account of the research process, findings, and recommendations, so that the other prospective researchers could evaluate the applicability to their context.

## Data analysis

Content data analysis was conducted using Tesch's method of the analysis process [24]. Transcribed interviews were read several times and descriptive words were identified for the formulation of thematic statements and concepts that created meaning. Notes were made with ideas that came to mind. Each interview was read through and the underlying meaning behind the scripts was determined. Notes were made on transcripts during the reading process. A list of concepts was made, and similar topics were clustered together. These clustered topics were then mapped as major topics. Raw data comprising transcripts were reviewed again in the light of identified topics. The text was coded and organised. Descriptive words were identified in the text and used to describe themes or topics. Related topics were grouped. A final decision regarding categories was made. Preliminary analysis was made and excerpts from the data supporting the identified categories and themes were coded and recoded. Due to virtual interviews, the researcher took field notes during each interview, based on the voice and body language. A reflective journal was used to form themes and sub-themes.

# Results

## Introduction

Data were collected from the purposively selected participants until data saturation was reached. Data analysis was conducted manually using the eight steps of Tesch's method of analysis. The participants' demographics, are illustrated in Table 1. The names of the participating hospitals and the participants were not divulged to protect their identities.

## Themes and subthemes

The views of some of the participants are presented as direct quotations to support the identified themes and sub-themes.

## Theme 1: Ineffective reporting system affecting communication of PSI guidelines

Communication factors are associated with the communication of PSI reporting and learning guidelines as perceived by healthcare professionals. Participants perceived communication to be ineffective and not user-friendly in implementing PSI reporting and learning guidelines. Most participants stated that the actual structure of the reporting system makes it difficult for healthcare professionals to effectively implement the guidelines.

**Subtheme 1.1: Tedious, long, and not simple.** The participants indicated that the reporting process was long and intense, tedious and laborious, hence other PSIs were not reported.

"*It is a long form. . .the new form is very intense, apart from the report that you're going to write. . .. not simple at all.*" *(ANM1)*

"*The time that it takes to fill in the PSI forms and then the possibility of further meetings down the line, it is essentially inconvenience from the interview process, a very laborious, inconvenient process.*" *(CMD1)*

**Table 1. Basic characteristics of participants.**

| Hospital | Gender | Job level |
|---|---|---|
| **FGD** | | OMs |
| Hospital A | 7 Females; 3 Males | |
| Hospital B | 8 Females; 2Males | |
| Hospitals C | 6 Females; 2Males | |
| **IDI** | | |
| Hospital A | 1 Female | ANMs, |
| Hospital B | 1 Male | |
| Hospitals C | 1 Female | |
| | | |
| Hospital A | 1Female; 2 Males | CMDs |
| Hospital B | 1 Female; 2 Males | |
| Hospitals C | 1 Male | |
| | | |
| Hospital A | 1 Female | M&E managers |
| Hospital B | 1 Male | |
| Hospitals C | 1 Female | |

**Subtheme 1.2: Unclear PSI guidelines and policies.** The healthcare professionals also stated that the guidelines were not clear, confusing, and not standardised. This was considered a barrier to the effective implementation of the guidelines.

"*The guidelines are not particularly clearer on when an adverse event can be considered as just expected as a consequence of the disease for a particular patient versus a true PSI. Our government is using their own terminology which is unfortunately not based on international standards.*" *(CMD2)*

Some participants verbalised that there was a mismatch of the guidelines with the actual work environment.

"*Action plans and policies are nice, and they are there, it's something that is there and are nice to read, but when it comes to practicality of it. . . . . . guidelines may look good on paper, but they are not effectively implemented.*" *(OM2)*

**Subtheme 1.3: Lack of consultation at the forefront level.** Healthcare professionals also reported a lack of consultation at the forefront level and nonclinical involvement during the development phase of PSI reporting and learning guidelines.

"*I was never consulted. . . I just received a directive from the DOH and my concern is that the people that develop these Guidelines are not clinical, they have no understanding of what is happening on the ground, they have no medical training.*" *(CMD2)*

"*I didn't have any input, maybe from other managers. I just received a new form from the province informing us what is required.*" *(ANM2)*

**Subtheme 1.4: Inadequate feedback to staff and family.** Inadequate feedback to staff and family was also identified by the participants as a barrier to the implementation of PSI reporting and learning guidelines. The lack of feedback was more from the senior management and was more of a redress meeting. Partial family involvement if a PSI has occurred, near misses are usually overlooked because it is perceived as harm that did not reach the patient.

"*So, people might say okay, what do we need to disclose that to the family because it was a near miss. It wasn't a miss. The patient didn't suffer any harm.*" *(CMD3)*

"*I've never had feedback. And I've done multiple incident reports reporting over the years, never, ever, ever, ever had. . .. you would expect documents and feedback. You don't ever get any feedback from them. It is a waste of time because you get nothing back. Then the next time something happens you go. . .. I'm not going to do it again because what was the point?*" *(CMD3)*

## Theme 2: Inadequate Institutional and management for the healthcare professionals

Most participants expressed a lack of support, especially from the senior management, and a disjuncture of the leadership from the challenges experienced by the frontline workers, resulting in poor teamwork.

**Subtheme 2.1: Punitive culture.**   Most participants reported that there is a punitive culture associated with the implementation of PSI reporting and learning guidelines. This was characterised by fear of the negative repercussions when healthcare professionals attempt to comply, therefore this led to underreporting of most PSIs.

"*Sometimes this leads to disciplinary measures. Sometimes you say, I'm going to give you a warning. and the PN will complain that I committed myself by reporting, and now I must face the negative consequences. They even threaten to consult the unions.*" **(ANM3)**

"*Also, people have a view that it is punitive, in a way. . . so, I have seen different departments threaten each other with PSIs. . . eh. . . so if you call something a PSI there is still very much a mindset that this is an accusation. . . and it is even been recognised between departments to sort of thresh out their politics. . ... . . been used between departments as a sort as a blame. Even various people are using it almost punitively. . .aggressively.*" **(CMD3)**

**Subtheme 2.2: A hostile environment.**   Participants described the environment as hostile, characterized by hierarchical structure and red tape resulting in reluctance in the implementation of PSI guidelines by the healthcare professionals.

"*Registered nurses that fear, they don't want to report that that's why they are under-reporting, because that fear you don't have support. It's always like a threatening environment. All they've learned is that they must keep quiet. And what if I get fired?*" **(CMD4)**

"*To me it's like your call to the court where you're gonna sit and when they discuss this incident you feel that I am. . . I am on the wrong. . .*" **(OM1)**

**Subtheme: 2.3 Lack of management support for the frontline healthcare workers.**   In addition, participants expressed a lack of guidance when trying to implement reporting and learning guidelines. Participants verbalised that an enabling environment is crucial, therefore inadequate organisational resources, patient safety committees that are not fully functional, lack of implementation strategies to guide healthcare professionals, and learning opportunities were regarded as barriers.

"*There is lack of support. Because. . . we are OMs, we work with staff, and we've got challenges. You need somebody to listen to you, if you say this and this is happening. Instead, you're blamed. . . . you are criticised. . . you. . . you. . . you really feel bad.*" **(OM3)**

"*No one's learning from the previous mistakes and there's always another excuse. You just hit your head against the wall.*" **(CMD5)**

Participants also verbalised that patient safety committees were not existing, and those that existed were at a development stage.

"*But from what I hear they just getting it up and running and getting some processes in order. I think the first steps are happening there. . . I view it as the start of the process. but I don't know if we've quite got the systems at the unit level. . . we don't have the unit SOP or patient safety committees on how to deal with a PSI. . . it is more of an ad-hoc base . . .*" **(CMD6)**

"*I'm not aware of any Patient Safety committee that exists. I don't even know what happens what the process is after I've handed the form as far as it goes from the ward to the ANM. . .*

*that's the point of the SOP. I don't know what happens after that, which is bad as well, because we need to know the entire process. . . it's all. . . it's all very haphazard". (CMD7)*

**2.4 Lack of teamwork.** Participants reiterated that there should be a learning experience during the implementation of reporting and learning guidelines, not to blame someone. Participants reported that a teamwork approach between departments and collaboration with other institutions was lacking.

"*We rely on nurses on the reporting of the PSI for formal reporting as such, because of a busy environment.*" *(CMD8).*

"*Although we morbidity and mortality meetings where we discuss the PSIs as staff, there is still that lack with collaboration with other institutions, we hardly meet. . .I don't remember us meeting. . ..so there is a gap there.*" *(ANM1)*

## Theme 3: Insufficient education and training of healthcare professionals

Participants reported that education and training associated with the implementation of PSI reporting and learning guidelines was multifaceted. The education and training challenges included the actual formal education of registered nurses, lack of training in specialisation of critical care, and inadequate in-service training and workshops, to empower the participants on the guidelines. Furthermore, The PSI guidelines were not made freely available and common knowledge to all the disciplines of departments.

**Subtheme 3.1: Lack of formal training of healthcare professionals.** Participants confirmed that some of the nurses that were allocated to the specialised units did not have the formal training and the skill that is required in this environment.

"*In most cases we were not even sure of their training course, as they were training in these schools . . .mushrooming everywhere, and now come in as ENAs and passed, they now want to be PNs, but still behaving like ENs. . .running the ward and there is no senior sister. You face these challenges, as they were the ENs recently and that affects the quality of nurses we employ.*" *(ANM2)*

"*People don't know how to identify PSIs are? How do you report. . .There's a total lack of education here. . .? So, you can't blame the individuals if they weren't educated in this. . . . . . and so I do think there's a major problem.*" *(CMD2)*

**Subtheme 3.2: Lack of knowledge on how to construct the report properly.** Furthermore, the participants indicated that the reporting system is not user-friendly with terminology that was not easy to understand. Although the participants knew about the classification of PSIs, the reporting process was complex.

"*I think the cause is a lack of knowledge of how to write a PSI. . . Lack of knowledge on how to construct properly. . . how to report properly, the whole incident. . . it has to be concise . . .it has to be to the point, so if it is not written properly, I send it back. . .so it's more time needed.*" *(ANM1)*

"*Then I mean, I'll say to the doctors you must fill out an incident reporting form and they'll go. . .What?. . . that nobody knows what it is. What to do with it.*" *(CMD1)*

**Subtheme 3.3: Lack of formal training, in-house training, and continued professional development programme.** Participants further indicated that there were insufficient continued professional development programmes to empower the healthcare professionals to construct the report properly. Participants are expected to have common knowledge for all healthcare professionals and to be familiar with the guidelines.

"*The perception that the staff is not adequately trained to identify the PSIs and to disclose to patients and relatives.*" *(ANM2)*

"*I think. . .. even the more senior clinician struggled to decide what is the PSI and what isn't a PSI. . .ah. . . so, do you label something a PSI*" *(CMD1)*

**Subtheme 3.4: Inadequate training of healthcare professionals on PSIs.** Some participants reported that some of the items that are in the PSI reporting and learning guidelines were well understood. They explained that some of the PSIs were expected as normal processes of the underlying illness.

"*Sometimes some of the items that are been included in those PSIs are not particularly appropriate to the ICU, because its normal to follow the normal pathology in ICU.*" *(CMD2)*

## Theme 4: Poor human resources affecting the implementation of PSI guidelines

Most participants confirmed that inadequate competent, knowledgeable, and experienced healthcare professionals were a huge barrier to the effective implementation of PSI reporting and learning guidelines. The inadequate staff allocation was exacerbated during the COVID-19 pandemic.

**4.1 Subtheme: Shortage of staff.** Specialised units are busy by nature, and normally have a ratio of 1:1, which is comprised of an adequate skills mix. Participants stated that it was impossible to achieve this ratio, therefore affecting the implementation of guidelines and compromising quality patient care.

"*The. . . conditions that we are working in are not allowing us to make sure can uhhh. . .. we try and ensure, but there isn't adequate staff that. . . staff is adequate, I mean people who are who work here every day, and who knows what is happening here and not people that I will be orientating. But the situation is that instead of having staff in this ward to work, according to the guidelines, or the policy we still get people from all over who will come here not knowing exactly what to do. Shortage itself is a big issue (emphatically)You get somebody that has just qualified as PN, she has never worked in a ward, she's just new. . .new PN She hasn't seen a ventilator she's seeing the ventilator for the first time. She's seeing the monitor for the first time.*" *(OM4)*

**Subtheme 4.2: Inadequate competent skill mix.** Participants also indicated that the staff that was allocated to their departments did not have specialised qualifications in SCU environment.

"*You need staff that will understand what is happening. . .. we don't even have time to orientate them. It's just orientation. . . .*" *(OM5)*

**Subtheme 4.3: High staff turnover and lack of staff retention.** Moreover, some participants expressed their frustration with high staff turnover and lack of staff retention.

"*There is also an increase in the number of retiring staff and the management is not recruiting for the vacant posts. . . they are not doing anything new. The staffing challenge is there. It takes too long to replace staff and you must motivate why you need more staff. Although there may be trained and experienced ICU nurses, they are not enough for the number of days and you find yourself not achieving that ratio of 1:1"* **(ANM3)**

**Subtheme 4.4: Busy environment.** Participants indicated that they work in a very busy environment, looking after vulnerable critically ill patients that require individual attention.

"*It is very difficult for staff, especially in this busy critical care where there are staff shortages. . . to now follow up . . . because somebody will be having a patient. . .remember here the ratio here is 1:1 or 1:2.*" **(ANM3)**

"*We work with sort of skeleton staff. Every day you wonder how people are going to cope with more admissions.*" **(OM6)**

## Discussion

This study sought to explore the views of the healthcare professionals, regarding the barriers and facilitators to the implementation of PSI reporting and learning guidelines in SCUs. Participants highlighted that although there are stipulated guidelines, they are not effectively executed by healthcare professionals. They reflected more barriers than facilitators which could be perceived as hindrances to the implementation of these guidelines. All participants acknowledged the importance of the revised improvement strategies to facilitate the rigorous implementation and improvement of quality patient care was essential.

### Theme 1: Ineffective reporting system affecting communication of PSI guidelines

In this study, the participants indicated that the difficulty was more on the process of communication of the PSI reporting and learning guidelines. This was a multi-level, amongst the individuals, departments, senior management, and the way the whole system is structured. Also, an organisational patient safety culture must be cultivated to create an atmosphere that is conducive to the communication of these guidelines. Barriers that were mostly mentioned by the participants included a laborious inconveniencing process, unclear guidelines and policies, not being simple and user-friendly, lack of consultation at the coalface, and lack of feedback. Several studies confirm that clear, standardised effective communication is key to effective implementation by healthcare professionals [1, 10, 11]. Although participants in Hospital B demonstrated a positive attitude toward the implementation of PSI reporting and learning guidelines, as they stated that they were partially involved during the consultative process, this was not the case in Hospital A and C, as they indicated that they were never consulted. The government plays a critical role in monitoring and evaluating the implemented PSI guidelines, therefore the lack of involvement of the Department of Health in managing incident reporting means that they are not aware of patient safety issues in the hospitals in their regions [28].

Lack of feedback to the staff and patients and uncertainty about what to report does not encourage the healthcare professionals to use the reporting system effectively, hence the

participants indicated that there were a lot of undocumented PSIs [3, 11, 15]. Participants also indicated that there is no learning that takes place, where healthcare professionals can use the PSI as a learning opportunity to prevent recurrences, as they learn from their mistakes. Healthcare professionals must not rely on reporting incidents only, as reports often do not lead to positive change [11]. Participants also stated that the feedback to patients and families is partial, only if it is a major event. Martos Algarra [29] affirms that patients become the first victims and the professionals involved turn into second victims and all of them need adequate support.

## Theme 2: Inadequate Institutional and management for the healthcare professionals

Investing in a patient safety culture can be viewed as building an organizational resource, which is beneficial for both, improving the care quality and protecting staff well-being [9]. There is a need for more effective transformational leadership to facilitate a culture of patient safety incident reporting and effective implementation of PSI reporting and learning guidelines [22]. In this study, participants explained an atmosphere that is hostile, and punitive and a lack of support of the frontline healthcare workers from the management [11, 22]. Fear of negative repercussions and litigation were the main barriers to the implementation of PSI reporting and learning guidelines [30]. According to Ontario et al., a non-punitive, confidential, independent, expert analysis, timely, system-oriented, and responsive was suggested [30, 31]. A hospital should particularly pay more attention to the confidentiality of case data in the system when establishing an incident reporting system, to avoid disputes and enhance reporting intention. Lack of teamwork and collaboration with other departments and institutions were also mentioned by the participants. A similar study highlighted the lack of collaboration between physicians and other healthcare personnel presented as barriers to improved patient safety due to perceived hierarchical differences [10]. Participants also stated that, although they had risk management meetings to discuss PSIs, there was a lack of guidance in the form of patient safety committees, implementation strategies, and standard operational procedures, that were of common knowledge. A similar study highlighted that the availability and use of written protocols that provide structure for their work, such as guidelines and standardized care plans, positively influence patient safety [10]. According to Dhamanti et al., the lack of commitment to, and priority of patient safety, the complexity of the bureaucratic structure, and a lack of systematic partnership and collaboration are problems that need to be addressed by systematic improvement [28].

## Theme 3: Insufficient education and training of healthcare professionals

Insufficient education and training, which was expressed by participants, was multifaceted. Participants mentioned poorly qualified healthcare professionals with no skills in working in a specialised care environment. Insufficient skills could act as a barrier to enhanced patient safety. Being a newly graduated nurse was mentioned as a barrier, due to a general lack of experience and the need for learning many new aspects of work [10]. Also, there was a lack of knowledge on how to identify PSIs, how to construct a written report, and adequate in-service training. A similar study confirms that the lack of knowledge about incident reporting systems, and lack of understanding about what constitutes an error were reported as common barriers [30]. The government plays a critical role in ensuring patient safety in healthcare services as they are actively involved in designing the guidelines and development of policies. Therefore, either the government or medical institute managers should collect and analyse the

information in the system, reduce the reoccurrence of medical errors through education and training, and improvement activities, and enhance the patient safety culture of the hospital [28].

### Theme 4: Poor human resources affecting the implementation of PSI guidelines

Participants reported the lack of human resources was not only the insufficient number of staff allocated to the specialised care units, but the skill mix was also lacking. The staff shortage was due to increased numbers of staff at retirement age, staff resigning for better employment, remuneration, and recruitment by other countries for competent staff. This brain drain could be associated with burnout syndrome and, therefore poor implementation of PSI reporting and learning guidelines. The specialised care units are characterised by a busy, stressful environment, with an increase in work overload and working with a skeleton staff that leads to burnout [32]. Shortage of staff and increased workload were associated with poor implementation of the PSI reporting guidelines [9, 16, 28]. Mossburg and Himmelfarb [32], affirm that healthcare managers can consider a simpler measure of this resource in ensuring adequate staffing levels across all departments of the institution.

### Strengths and limitations of the study

This study provided thought-provoking findings as in-depth information was collected from the focus group discussions and individual interviews with participants who were in managerial positions and were actively involved in the implementation of the PSI reporting and learning guidelines. In addition, the purposive sample from the focus group was homogenous, whereas the individual interviews were heterogenous, which provided a rich array of experiences from various healthcare professionals. To the researchers' knowledge, this is the first study to explore the barriers to the implementation of PSI reporting and learning guidelines by healthcare professionals in South African SCUs. Furthermore, the study findings may inform the policymakers and healthcare facilities to revise the PSI reporting guidelines, to be more relevant to the South African context and devise better implementation strategies, to improve patient safety care. In addition, the findings from this study have identified areas that need improvement in patient safety, therefore its dissemination contributes to the cultivation of patient safety culture worldwide. The study participants were limited to three major government hospitals in KwaZulu-Natal; however, the sample was representative enough to address the research question. Also, most of the literature that supported this study is from international sources indicating a paucity of South African literature on this topic.

### Recommendations

This study was conducted in one province; therefore, it is recommended that further research be conducted in other provinces. The study analysis and results can be used by national and global stakeholders to improve the implementation of PSI reporting and learning guidelines, as per the WHO recommendations. For rigorous implementation in South African SCUs, the study recommends revised PSI reporting and learning guidelines that consist of standardised, simple -user-friendly terminology as well as a better implementation strategy to guide the healthcare professionals to apply the guidelines effectively. These guidelines need to be designed in consultation with frontline healthcare professionals and policymakers. Continuous professional development programmes may play an important role in facilitating the implementation process. Education on PSI reporting and learning guidelines should be

incorporated early in the curriculum of healthcare professionals so that it becomes common knowledge for all.

## Conclusion

This study confirmed that PSI reporting and learning guidelines are still not effectively implemented in the SCUs and the barriers to implementation were highlighted. There is a need to implement these guidelines globally. The study revealed that ineffective reporting system, lack of institutional and managerial support, insufficient education and training of healthcare professionals, and poor human resources were the main barriers to the effective implementation of PSI and learning guidelines. To effectively improve patient safety, this study seeks to strengthen the collaboration among healthcare professionals, and organisations, within the African continent and globally.

## Supporting information

**S1 Appendix. Interview guide for the focus group.**
(PDF)

**S2 Appendix. Individual interview guide.**
(PDF)

**S3 Appendix. Data set-minimal.**
(DOCX)

## Acknowledgments

The authors would like to acknowledge the gatekeepers and the healthcare professionals from the three hospitals who assisted and participated in the study.

## Author Contributions

**Conceptualization:** T. M. H. Gqaleni, Sipho W. Mkhize.

**Data curation:** T. M. H. Gqaleni.

**Formal analysis:** T. M. H. Gqaleni.

**Funding acquisition:** T. M. H. Gqaleni.

**Investigation:** T. M. H. Gqaleni.

**Methodology:** T. M. H. Gqaleni.

**Project administration:** T. M. H. Gqaleni.

**Resources:** T. M. H. Gqaleni.

**Software:** T. M. H. Gqaleni.

**Supervision:** Sipho W. Mkhize.

**Validation:** Sipho W. Mkhize.

**Visualization:** Sipho W. Mkhize.

**Writing – original draft:** T. M. H. Gqaleni.

**Writing – review & editing:** T. M. H. Gqaleni, Sipho W. Mkhize.

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
