## [Decision Letter · Decision Letter 0]

19 Sep 2023

PONE-D-23-22765Barriers influencing implementation of patient safety incident reporting and learning guidelines in specialised care units, KwaZulu-Natal: A qualitative study.PLOS ONE

Dear Dr. Gqaleni,

Thank you for submitting your manuscript to PLOS ONE. After careful consideration, we feel that it has merit but does not fully meet PLOS ONE’s publication criteria as it currently stands. Therefore, we invite you to submit a revised version of the manuscript that addresses the points raised during the review process.

We look forward to receiving your revised manuscript.

Kind regards,

Bereket Yakob, Ph.D.

Academic Editor

PLOS ONE

Journal Requirements:

5. We note you have included a table to which you do not refer in the text of your manuscript. Please ensure that you refer to Table 1 in your text; if accepted, production will need this reference to link the reader to the Table.

Reviewers' comments:

Reviewer's Responses to Questions

**Comments to the Author**

1. Is the manuscript technically sound, and do the data support the conclusions?

Reviewer #1: Yes

Reviewer #2: Yes

Reviewer #3: Yes

2. Has the statistical analysis been performed appropriately and rigorously? 

Reviewer #1: Yes

Reviewer #2: N/A

Reviewer #3: Yes

3. Have the authors made all data underlying the findings in their manuscript fully available?

Reviewer #1: Yes

Reviewer #2: Yes

Reviewer #3: Yes

4. Is the manuscript presented in an intelligible fashion and written in standard English?

Reviewer #1: Yes

Reviewer #2: Yes

Reviewer #3: Yes

5. Review Comments to the Author

Reviewer #1: Can you please provide a explanation of the published article 'Gqaleni TM, Mkhize SW. Healthcare professionals’ perception of knowledge and implementation of Patient Safety Incident reporting and learning guidelines in specialised care units, KwaZulu-Natal. Southern African Journal of Critical Care. 2023 Mar 1;39(1):25-9.' and how this submission adds to what is already been presented in the article cited.

Reviewer #2: The manuscript is well written and have scientific evidence, the results are well presented, and discussion of results is provided in a logic way and supported by literature. The methodology needs to be described in full, how participants were recruited and consented to participate virtually. Due to lockdown restriction how were interviews done, which virtual platforms were used and how were ethical issues handled with these platforms. Again, field notes and observations during interviews. The population was highlighted but the anticipated sample size was not indicated, when did data saturation occur? The prolonged engagement is not clear, was it determined by time the duration of the interviews?

Reviewer #3: Thank you for inviting me to review this valuable work. The authors have done great work, from conceptualizing the topic to writing the manuscript. Below are some comments and feedback for your consideration:

1. In the abstract, the study timeframe was stated as from March to May 2021; however, line 123 mentions it as from August to October 2021. Please revise and adjust this inconsistency accordingly.

2. In line 58, please add the reference number at the end of the sentence.

3. Please review the entire manuscript and use either parentheses ( ) or brackets [ ] consistently for in-text reference citations.

4. Consider adding one or two sentences about Table 1 after line 110 and insert the table just before the recruitment strategy section.

5. Lines 102 and 166-186 contain extra information that does not appear to be necessary for the manuscript.

6. Please clarify why participants provided verbal consent instead of written, electronic consent.

7. While the interviews were conducted virtually, specifying which platform was used would be beneficial.

8. Label Table 2 as "Basic Characteristics of Participants" and, if available, include the mean age and years of experience.

9. For Table 3, please describe the themes in the text or the table, but not both.

10. In lines 477 and 490, mention only the first author and add "et al."

11. In the conclusion, consider removing "middle and low-income countries."

6. PLOS authors have the option to publish the peer review history of their article (what does this mean?). If published, this will include your full peer review and any attached files.

Reviewer #1: No

Reviewer #2: No

Reviewer #3: **Yes: **Mona Abdelrehim

---

## [Author Response · Author response to Decision Letter 0]

3 Nov 2023

Thank you for the invitation to submit a revised version of the manuscript that addresses the points raised during the review process. The comments were constructive and assisted me to revise and improve the quality of the manuscript.

The funding is from South African National Research foundation, but I do not t find it in your drop-down menu (https://www.nrf.ac.za/)

---

## [Decision Letter · Decision Letter 1]

8 Dec 2023

PONE-D-23-22765R1Barriers influencing implementation of patient safety incident reporting and learning guidelines in specialised care units, KwaZulu-Natal: A qualitative study.PLOS ONE

Dear Dr. Gqaleni,

Thank you for submitting your manuscript to PLOS ONE. After careful consideration, we feel that it has merit but does not fully meet PLOS ONE’s publication criteria as it currently stands. Therefore, we invite you to submit a revised version of the manuscript that addresses the points raised during the review process.

Please submit your revised manuscript by Jan 22 2024 11:59PM. If you will need more time than this to complete your revisions, please reply to this message or contact the journal office at plosone@plos.org. Please include the following items when submitting your revised manuscript:A rebuttal letter that responds to each point raised by the academic editor and reviewer(s). You should upload this letter as a separate file labeled 'Response to Reviewers'.A marked-up copy of your manuscript that highlights changes made to the original version. You should upload this as a separate file labeled 'Revised Manuscript with Track Changes'.An unmarked version of your revised paper without tracked changes. You should upload this as a separate file labeled 'Manuscript'.If applicable, we recommend that you deposit your laboratory protocols in protocols.io to enhance the reproducibility of your results. Protocols.io assigns your protocol its own identifier (DOI) so that it can be cited independently in the future. For instructions see: https://journals.plos.org/plosone/s/submission-guidelines#loc-laboratory-protocols. Additionally, PLOS ONE offers an option for publishing peer-reviewed Lab Protocol articles, which describe protocols hosted on protocols.io. Read more information on sharing protocols at https://plos.org/protocols?utm_medium=editorial-email&utm_source=authorletters&utm_campaign=protocols.

We look forward to receiving your revised manuscript.

Kind regards,

Bereket Yakob, Ph.D.

Academic Editor

PLOS ONE

Journal Requirements:

**Additional Editor Comments:**

Title: It can be shortened to “Barriers to implementing patient safety incident reporting and learning guidelines in specialised care units, KwaZulu-Natal: A qualitative study”

Rephrase “senior healthcare workers” into “healthcare workers in specialized care units.” The use of “senior” is misleading as it was not defined. If you insist on keeping the term, define it and insert the criteria you used.

Remove the “exclusion criteria.” These were simply the reverses of the inclusion criteria. The participants cannot be included if they do not meet the inclusion criteria. If no other criteria were used to exclude included participants, stating the exclusion criteria is unnecessary.

Reviewers' comments:

Reviewer's Responses to Questions

**Comments to the Author**

1. If the authors have adequately addressed your comments raised in a previous round of review and you feel that this manuscript is now acceptable for publication, you may indicate that here to bypass the “Comments to the Author” section, enter your conflict of interest statement in the “Confidential to Editor” section, and submit your "Accept" recommendation.

Reviewer #3: All comments have been addressed

Reviewer #4: (No Response)

2. Is the manuscript technically sound, and do the data support the conclusions?

Reviewer #3: Yes

Reviewer #4: Yes

3. Has the statistical analysis been performed appropriately and rigorously? 

Reviewer #3: Yes

Reviewer #4: N/A

4. Have the authors made all data underlying the findings in their manuscript fully available?

Reviewer #3: Yes

Reviewer #4: No

5. Is the manuscript presented in an intelligible fashion and written in standard English?

Reviewer #3: Yes

Reviewer #4: Yes

6. Review Comments to the Author

Reviewer #3: The authors have addressed all the comments properly. The only concern is that Table 2 has extra numbers not related to the table itself. Please revise accordingly.

Reviewer #4: Methods

- Line 133 The rationale for only including senior members of the PSI reporting team should be included. For example, junior members may be more honest about PSI reporting issues as they are not in management and less likely to cover up.

- Line 156 Data collection process should describe the topic guide used - was a conceptual framework used? Or a list of topics explored should be given. Language and translation process should also be described if non-English language was used.

- Line 211 Trustworthiness- Suggest to remove this section and any elements felt important can be embedded in the methodology sections above.

- Table 1 can be revised to remove the exclusion criteria column, as reversing the inclusion is not useful.

Results

- Line 254-258 , these abbreviations should be included in the text at first use, or embedded into Table 2.

- In reporting quotes, suggest to remove whether it was an FGD or IDI, as it is known that OMs are FGDs and also it is not necessary to know if the quote came from FGD or IDI.

Otherwise it is noted that the author has made revisions based on prior reviewer comments which have been adequately addressed.

7. PLOS authors have the option to publish the peer review history of their article (what does this mean?). If published, this will include your full peer review and any attached files.

Reviewer #3: **Yes: **Mona Abdelrehim

Reviewer #4: No

---

## [Author Response · Author response to Decision Letter 1]

24 Jan 2024

Dear Editor,

Thank you for the comments that have assisted me to improve the quality of this manuscript. Minor revisions were made. 

I was not aware that I had to re-upload the supporting information as I uploaded it with the first submission.

Kind Regards,

Mrs Gqaleni

---

## [Editor Report · Decision Letter 2]

6 Feb 2024

Barriers to implementing patient safety incident reporting and learning guidelines in specialised care units, KwaZulu-Natal: A qualitative study.

PONE-D-23-22765R2

Dear Dr. Gqaleni,

We’re pleased to inform you that your manuscript has been judged scientifically suitable for publication and will be formally accepted for publication once it meets all outstanding technical requirements.

Kind regards,

Bereket Yakob, Ph.D.

Academic Editor

PLOS ONE

Additional Editor Comments (optional):

Please avoid unnecessary use of block letters at the beginning of certain words in the middle of the text, for example Patient Safety Incident, Guidelines, Chief Executive Officers, Critical Care Units, Operational Nurse Managers, Assisting Nurse Managers, Monitoring, and Evaluation Managers, and Consultant, Medical Doctors, etc.
---

## [Editor Report · Acceptance letter]

25 Feb 2024

PONE-D-23-22765R2 

PLOS ONE

Dear Dr. Gqaleni, 

I'm pleased to inform you that your manuscript has been deemed suitable for publication in PLOS ONE. Congratulations! Your manuscript is now being handed over to our production team.

Kind regards, 

on behalf of

Dr. Bereket Yakob 

Academic Editor

PLOS ONE